# Information Theory for Biological Sequence Classification: A Novel Feature Extraction Technique Based on Tsallis Entropy

**DOI:** 10.3390/e24101398

**Published:** 2022-10-01

**Authors:** Robson P. Bonidia, Anderson P. Avila Santos, Breno L. S. de Almeida, Peter F. Stadler, Ulisses Nunes da Rocha, Danilo S. Sanches, André C. P. L. F. de Carvalho

**Affiliations:** 1Institute of Mathematics and Computer Sciences, University of São Paulo, São Carlos 13566-590, Brazil; 2Department of Environmental Microbiology, Helmholtz Centre for Environmental Research-UFZ GmbH, 04318 Leipzig, Germany; 3Department of Computer Science and Interdisciplinary Center of Bioinformatics, University of Leipzig, 04107 Leipzig, Germany; 4Department of Computer Science, Federal University of Technology-Paraná—UTFPR, Cornélio Procópio 86300-000, Brazil

**Keywords:** feature extraction, tsallis entropy, biological sequence, information theory

## Abstract

In recent years, there has been an exponential growth in sequencing projects due to accelerated technological advances, leading to a significant increase in the amount of data and resulting in new challenges for biological sequence analysis. Consequently, the use of techniques capable of analyzing large amounts of data has been explored, such as machine learning (ML) algorithms. ML algorithms are being used to analyze and classify biological sequences, despite the intrinsic difficulty in extracting and finding representative biological sequence methods suitable for them. Thereby, extracting numerical features to represent sequences makes it statistically feasible to use universal concepts from Information Theory, such as Tsallis and Shannon entropy. In this study, we propose a novel Tsallis entropy-based feature extractor to provide useful information to classify biological sequences. To assess its relevance, we prepared five case studies: (1) an analysis of the entropic index *q*; (2) performance testing of the best entropic indices on new datasets; (3) a comparison made with Shannon entropy and (4) generalized entropies; (5) an investigation of the Tsallis entropy in the context of dimensionality reduction. As a result, our proposal proved to be effective, being superior to Shannon entropy and robust in terms of generalization, and also potentially representative for collecting information in fewer dimensions compared with methods such as Singular Value Decomposition and Uniform Manifold Approximation and Projection.

## 1. Introduction

The accelerated evolution of sequencing technologies has generated significant growth in the number of sequence data [1], opening up new opportunities and creating new challenges for biological sequence analysis. To take advantage of the increased predictive power of machine learning (ML) algorithms, recent works have investigated the use of these algorithms to analyze biological data [2,3].

The development of effective methods for sequence analysis, through ML, benefits the research advancement in new applications [4,5], such as understanding several problems [4,5], e.g., cancer diagnostics [6], development of CRISPR-Cas systems [7], drug discovery and development [8] and COVID-19 diagnosis [9]. Nevertheless, ML algorithms applied to the analysis of biological sequences present challenges, such as feature extraction [10]. For non-structured data, as is the case of biological sequences, feature extraction is a key step for the success of ML applications [11,12,13].

Previous works have shown that universal concepts from Information Theory (IT), originally proposed by Claude Shannon (1948) [14], can be used to extract relevant information from biological sequences [15,16,17]. According to [18], an IT-based analysis of symbolic sequences is of interest in various study areas, such as linguistics, biological sequence analysis or image processing, whose relevant information can be extracted, for example, by Shannon’s uncertainty theory [19].

Studies have investigated the analysis of biological sequences with Shannon entropy in a wide range of applications [19,20,21]. Given their large applicability, according to [22], it is important to explore the possibility of generalized entropies, such as Tsallis [23,24], which was proposed to generalize the Boltzmann/Gibbs’s traditional entropy to non-extensive physical systems [25]. This class of generalized entropy has been used for different problems, e.g., image analysis [25,26], inference of gene regulatory networks [27], DNA analysis [20] induction of decision trees [28] and classification of epileptic seizures [29].

In [25], the authors proposed a new image segmentation method using Tsallis entropy. Later, Ref. [26] showed a novel numerical approach to calculate the Tsallis entropic index feature for a given image. In [27], the authors introduced the use of generalized entropy for the inference of gene regulatory networks. DNA analysis using entropy (Shannon, Rényi and Tsallis) and phase plane concepts was presented in [20], while [28] used the concept of generalized entropy for decision trees. Recently, Ref. [29] investigated a novel single feature based on Tsallis entropy to classify epileptic seizures. These studies report a wide range of contributions to the use of Tsallis entropy in different domains. To the best of our knowledge, this paper is the first work proposing its use as a feature (feature extraction) to represent distinct biological sequences. Additionally, it presents the first study of different Tsallis entropic indexes and their effects on classical classifiers.

A preliminary version of this proposal was presented in [5]. Due to the favorable results obtained, we created a code to extract different descriptors available in a new programming package, called MathFeature [13], which implements mathematical descriptors for biological sequences. However, until now, we have not studied Tsallis entropy in depth, e.g., its effect, its application to other biological sequence datasets and its comparison with other entropy-based descriptors, e.g., Shannon. Thus, in this paper, we investigate the answers to the following questions:**Question 1 (Q1):** Are Tsallis entropy-based features robust for extracting information from biological sequences in classification problems?**Question 2 (Q2):** Does the entropic index affect the classification performance?**Question 3 (Q3):** Is Tsallis entropy as robust as Shannon entropy for extracting information from biological sequences?

We are evaluating robustness in terms of performance, e.g., accuracy, recall and F1 score, of the feature vectors extracted by our proposal on different biological sequence datasets. Finally, this study makes the following main research contributions: we propose an effective feature extraction technique based on Tsallis entropy, being robust in terms of generalization, and also potentially representative for collecting information in fewer dimensions for sequence classification problems.

## 2. Literature Review

In this section, we develop a systematic literature review to present and summarize feature extraction descriptors for biological sequences (DNA, RNA, or protein). This review aims to report the need and lack of studies with mathematical descriptors, such as entropy, evidencing the contribution of this article. This section followed the Systematic Literature Review (SLR) Guidelines in Software Engineering [30], which, according to [30,31], allows a rigorous and reliable evaluation of primary studies within a specific topic. We base our review on recommendations from previous studies [30,31,32].

We propose to address the following problem: *How can we numerically represent a biological sequence (such as DNA, RNA, or protein) in a numeric vector that can effectively reflect the most discriminating information in a sequence?* To answer this question, we reviewed ML-based feature extraction tools (or packages, web servers and toolkits) that aim, as a proposal, to provide several feature descriptors for biological sequences—that is, without a defined scope, and, therefore, generalist studies. Moreover, we used the following electronic databases: ACM Digital Library, IEEE Xplore Digital Library, PubMed and Scopus. We chose the Boolean method [33] to search primary studies in the literature databases. The standard search string was: *(“feature extraction” OR “extraction” OR “features” OR “feature generation” OR “feature vectors”) AND (“machine” OR “learning”) AND (“tool” OR “web server” OR “package” OR “toolkit”) AND (“biological sequence” OR “sequence”).*

Due to different query languages and limitations between the scientific article databases, there were some differences in the search strings. Therefore, our first step was to apply search keys to all databases, returning a set of 1404 studies. Furthermore, we used the Parsifal tool to assist our review and obtain better accuracy and reliability. Thereafter, duplicate studies were removed, returning an amount of 1097 titles (307 duplicate studies).

Then, we performed a thorough analysis of the titles, keywords and abstracts, according to **inclusion and exclusion criteria**: (1) Studies in English, (2) Studies with different feature extraction techniques, (3) Studies with generalist tools and (4) Studies published in journals. We accepted 28 studies (we rejected, 1069). Finally, after pre-selecting the studies, we performed a data synthesis, to apply an assessment based on the **quality criteria**: (1) Are the study aims specified? (2) Study with different proposals/results? (3) Study with complete results?

Hence, of the 28 studies, 3 were eliminated, leading to a final set of 25 studies (see Appendix A). As previously mentioned, we assessed generalist tools for feature extraction, since this type of study would provide several descriptors, presenting an overview of ways to numerically represent biological sequences (which would not be possible by evaluating studies dedicated to some specific problem). As expected, we found more than 100 feature descriptors. We chose to divide them into large groups (16 groups—these were defined based on all studies), as shown in Appendix A. Then, we created Table 1 with all the feature descriptors found in the 25 studies (see the complete table in Appendix A). As can be seen, no study provides mathematical descriptors, such as Tsallis entropy, reinforcing the contribution of our proposal.

## 3. Information Theory and Entropy

According to [34], IT can be defined as a mathematical treatment of the concepts, parameters, and rules related to the transmission and processing of information. The IT concept was first proposed by Claude Shannon (1948) in the work entitled “A Mathematical Theory of Communication” [14], where he showed how information could be quantified with absolute precision. The entropy originating from IT can be considered a measure of order and disorder in a dynamic system [14,25]. However, to define information and entropy, it is necessary to understand *random variables*, which, in probability theory, is a mathematical object that can take on a finite number of different states $x_{1},\ldots ,x_{n}$ with previously defined probabilities $p_{1},\ldots ,p_{n}$ [35]. According to [5], for a discrete random variable *R* taking values in {r[0],r[1],r[2],…,r[N−1]} with probabilities {p[0],p[1],p[2],…,p[N−1]}, represented as P(R=r[n])=p[n], we can define self-information or information as [26]
(1)I=−log(p).

Thus, the Shannon entropy $H_{S}$ is defined by
(2)$$H_{S}=-\sum_{n=0}^{N-1}p\left[n\right]\;log_{2}\;p\left[n\right].$$

Here, *N* is the number of possible events and p[n] the probability that event *n* occurs. Fundamentally, with Shannon entropy, we can reach a single value that quantifies the information contained in different observation periods [36]. Furthermore, it is important to highlight that the Boltzmann/Gibbs entropy was redefined by Shannon as a measure of uncertainty [25]. This formalism, known as Boltzmann–Gibbs–Shannon (BGS) statistics, has often been used to interpret discrete and symbolic data [18]. Moreover, according to [25,37], if we decompose a physical system into two independent statistical subsystems A and B, the Shannon entropy has the extensive property (additivity)
(3)$$H_{S}\left(A+B\right)=H_{S}\left(A\right)+H_{S}\left(B\right)$$

According to [38], complementary information on the importance of specific events can be generated using the notion of generalized entropy, e.g., outliers or rare events. Along these lines, Constantino Tsallis [23,24] proposed a generalized entropy of the BGS statistics, which can be defined as follows:(4)$$H_{T}=\frac{1}{q-1}\left(1-\sum_{n=0}^{N-1}p{\left[n\right]}^{q}\right).$$

Here, *q* is called the entropic index, which, depending on its value, can represent various types of entropy. Depending on the value of *q*, three different entropies can be defined [25,37]:Superextensive entropy (q<1):
(5)$$H_{T}\left(A+B\right)<H_{T}\left(A\right)+H_{T}\left(B\right)$$Extensive entropy (q=1):
(6)$$H_{T}\left(A+B\right)=H_{T}\left(A\right)+H_{T}\left(B\right)$$Subextensive entropy (q>1):
(7)$$H_{T}\left(A+B\right)>H_{T}\left(A\right)+H_{T}\left(B\right)$$

When q<1, the Tsallis entropy is superextensive; for q=1, it is extensive (e.g., leads to the Shannon entropy), and for q>1, it is subextensive [39]. Therefore, based on these differences, it is important to explore the possibility of generalized entropies [22,28,40]. Another notable generalized entropy is the Rényi entropy, which generalizes the Shannon entropy, the Hartley entropy, the collision entropy and the min-entropy [41,42]. The Rényi entropy can be defined as follows:(8)$$H_{R}=\frac{1}{1-q}log_{2}\left(\sum_{n=0}^{N-1}p{\left[n\right]}^{q}\right).$$

As in the Tsallis entropy, q=1 leads to Shannon entropy.

## 4. Materials and Methods

In this section, we describe the experimental methodology adopted for this study, which is divided into five stages: (1) data selection; (2) feature extraction; (3) extensive analysis of the entropic index; (4) performance analysis; (5) comparative study.

### 4.1. A Novel Feature Extraction Technique

Our proposal is based on the studies of [5,20]. To generate our probabilistic experiment [15], we use a known tool in biology, the k-mer. In this method, each sequence is mapped in the frequency of neighboring bases *k*, generating statistical information. The k-mer is denoted in this work by $P_{k}$, corresponding to Equation (Equation 9).
(9)$$\begin{array}{cc} P_{k}\left(s\right)=\frac{c_{i}^{k}}{N-k+1}=(\; & \frac{c_{1}^{1}}{N-1+1},\ldots ,\frac{c_{4}^{1}}{N-1+1},\\ \; & \frac{c_{4+1}^{2}}{N-2+1},\ldots ,\frac{c_{i}^{k}}{N-k+1})\;k=1,2,\ldots ,n. \end{array}$$

Here, each sequence (s) was assessed with frequencies of k=1,2,…,24, in which $c_{i}^{k}$ is the number of occurrences with length *k* in a sequence (s) with length *N*; the index $i\in \{1,2,\ldots ,4^{1}+\ldots +4^{k}\}$ refers to an analyzed substring (e.g., [{AAAA},…,{TTTT}], for k=4). Here, after counting the absolute frequencies of each *k*, we generate relative frequencies and then apply Tsallis entropy to generate the features. In the case of protein sequences, index *i* is $\left\{1,2,\ldots ,20^{1}+\ldots +20^{k}\right\}$. For a better understanding, Algorithm 1 demonstrates our pseudocode.
**Algorithm 1:** Pseudocode of the Proposed Technique
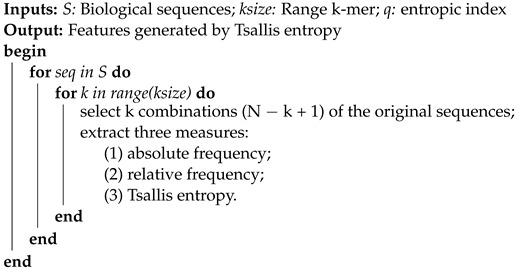


This algorithm is divided into five steps: (1) each sequence is mapped to k−mers; (2) extraction of the absolute frequency of each k−mer; (3) extraction of the relative frequency of each k−mer based on absolute frequency; (4) extraction of the Tsallis entropy, based on the relative frequency for each k−mer—see Equation (Equation 4); (5) generation, for each k−mer, of an entropic measure. Regarding interpretability, each entropic measure represents a k−mer, e.g., 1-mer = frequency of A, C, T, G. In other words, analyzing the best measures—for example, through a feature importance analysis—we can determine which k−mers are more relevant to the problem under study, providing an indication of which combination of nucleotides or amino acids contributes to the classification of the sequences.

### 4.2. Benchmark Dataset and Experimental Setting

To validate the proposal, we divided our experiments into five case studies:**Case Study I:** Assessment of the Tsallis entropy and the effect of the entropic index *q*, generating 100 feature vectors for each benchmark dataset with 100 different *q* parameters (entropic index). The features were extracted by Algorithm 1, with *q* varying from 0.1 to 10.0 in steps of 0.1 (except 1.0, which leads to the Shannon entropy). The goal was to find the best values for the parameter *q* to be used in the experiments. For this, three benchmark datasets from previous studies were used [5,43,44]. For the first dataset (D1), the selected task was long non-coding RNAs (lncRNA) vs. protein-coding genes (mRNA), as in [45], using a set with mRNA and lncRNA sequences (500 for each label—benchmark dataset [5]). For the second dataset (D2), a benchmark set from [5], the selected task was the induction of a classifier to distinguish circular RNAs (cirRNAs) from other lncRNAs using 1000 sequences (500 for each label). The third dataset (D3) is for Phage Virion Protein (PVP) classification, from [44], with 129 PVP and 272 non-PVP sequences.**Case Study II:** We use the best parameters (q: entropic index—found in case study I) to evaluate its performance on new datasets: D4—Sigma70 Promoters [46] (2141 sequences), D5—Anticancer Peptides [47] (344 sequences) and D6—Severe Acute Respiratory Syndrome Coronavirus 2 (SARS-CoV-2, 24815 sequences) [13].**Case Study III—Comparing Tsallis with Shannon Entropy:** As a baseline of the comparison between methods, we use Shannon entropy, as we did not find any article studying the form of proposed classification with Tsallis entropy and the effect of the entropic parameter with different classifiers. In this experiment, we use D1, D2, D3, D4, D5 and D6.**Case Study IV—Comparing Generalized Entropies:** To better understand the effectiveness of generalized entropies for feature extraction, we evaluated Tsallis with the Rényi entropy. In this case, the evaluations of the two approaches were conducted by using the experiments from case study I, changing the entropic index for generating the datasets from 0.1 to 10.0 in steps of 0.1, and inducing the CatBoost classifier. In addition, the datasets used were D1, D2 and D3.**Case Study V—Dimensionality Reduction Analysis:** Finally, we assessed our proposal with other known techniques of feature extraction and dimensionality reduction, e.g., Singular Value Decomposition (SVD) [48] and Uniform Manifold Approximation and Projection (UMAP) [49], using datasets D1, D2, D3 and D5. We also added three new benchmark datasets provided by [50] to predict recombination spots (D7) with 1050 sequences (it contained 478 positive sequences and 572 negative sequences) and for the HIV-1 M pure subtype against CRF classification (D8) with 200 sequences (it contained 100 positive and negative sequences) [51]. In addition, we also used a multiclass dataset (D9) containing seven bacterial phyla with 488 small RNA (sRNA), 595 transfer RNA (tRNA) and 247 ribosomal RNA (rRNA) from [52]. Moreover, to apply SVD and UMAP, we kept the same feature descriptor by k-mer frequency.

For data normalization in all stages, we used the min–max algorithm. Furthermore, we investigated five classification algorithms, such as Gaussian Naive Bayes (GaussianNB), Random Forest (RF), Bagging, Multi-Layer Perceptron (MLP) and CatBoost. To induce our models, we randomly divided the datasets into ten separate sets to perform 10-fold cross-validation (case study I and case study V) and hold-out (70% of samples for training and 30% for testing—case study II, case study III, and case study IV). Finally, we assessed the results with accuracy (ACC), balanced accuracy (BACC), recall, F1 score and Area Under the Curve (AUC). In D9, we considered metrics suitable for multiclass evaluation.

## 5. Results and Discussion

### 5.1. Case Study I

As aforementioned, we induced our classifiers (using 10-fold cross-validation) across all feature vectors generated with 100 different *q* parameters (totaling 300 vectors (3 datasets times 100 parameters)). Thereby, we obtained the results presented in Table 2. This table shows the best and worst parameter (entropic parameter *q*) of each algorithm in the three benchmark datasets, taking into account the ACC metric.

Thereby, evaluating each classifier, we observed that the CatBoost performed best in all datasets, with 0.9440 (q=2.3), 0.8300 (q=4.0), 0.7282 (q=1.1) in D1, D2 and D3, respectively. The other best classifiers were RF, with 0.9430 (*q* = 0.4 − D1) and 0.8220 (*q* = 5.3 − D2), followed by Bagging, MLP, and GaussianNB. Furthermore, in general, we noticed that the best results presented parameters between 1.1<q<5.0, i.e., when the Tsallis entropy was subextensive. Along the same lines, it can be observed in Table 2 that the worst parameters are between 9.0<q<10.0, when the Tsallis entropy is also subextensive. However, for a more reliable analysis, we plotted graphs with the results of all tested parameters (0.1 to 10.0 in steps of 0.1), as shown in Figure 1.

A large difference can be observed in the entropy obtained by each parameter *q*, mainly in benchmark D3. Thereby, analyzing D1 and D2, we noticed a pattern of robust results until q=6, for the best classifiers in both datasets. However, as the *q* parameter increases, the classifiers are less accurate. On the other hand, if we look at D3, the entropy obtained for each parameter *q* presents a much greater variation, but following the same drop with parameters close to q=10. Regarding the superextensive entropy (q<1), some cases showed robust results; however, most classifiers behaved better with the subextensive entropy.

### 5.2. Case Study II

After substantially evaluating the entropic index, our findings indicated that the best parameters were among 1.1<q<5.0. Thereby, we generated new experiments using five parameters to test their efficiency in new datasets, with q=(0.5,2.0,3.0,4.0,5.0), as shown in Table 3 (sigma70 promoters—D4), Table 4 (anticancer peptides—D5) and Table 5 (SARS-CoV-2—D6). Here, we generated the results with the two best classifiers (RF and Catboost—best in bold).

Assessing each benchmark dataset, we note that the best results were of ACC: 0.6687 and AUC: 0.6108 in D4 (RF, q=2.0), ACC: 0.7212 and AUC: 0.7748 in D5 (RF, q=3.0), and ACC: 1.0000 and AUC: 1.0000 in D5 (RF and CatBoost, q=5.0). Once more, the results confirm that the best parameters are in the range of 1.1<q<5.0, indicating a good choice when using Tsallis entropy. The perfect classification at D6 is supported by other studies in the literature [53,54,55]. Nevertheless, after testing the Tsallis entropy on six benchmark datasets, we noticed an indication that this approach behaves better with longer sequences, e.g., D1 (mean length ≈ 751 bp), D2 (mean length ≈ 2799 bp), and D6 (mean length ≈ 10,870 bp) showed robust results, while D3 (mean length ≈ 268 bp), D4 (mean length ≈ 81 bp), and D5 (mean length ≈ 26 bp) showed less accurate results. Nonetheless, Tsallis entropy could contribute to hybrid approaches, as our proposal achieved relevant results in four datasets.

### 5.3. Case Study III—Comparing Tsallis with Shannon Entropy

Here, we used Shannon entropy as a baseline for comparison, according to Table 6. Various studies have covered the biological sequence analysis with Shannon entropy, in the most diverse applications. For a fair analysis, we reran the experiments on all datasets (case study I and II, six datasets), using hold-out, with the same train and test partition for both approaches. Once more, we used the best classifiers in case study II (RF and CatBoost), but, for a better understanding, we only show the best result in each dataset.

According to Table 6, our proposal with Tsallis entropy showed better results of ACC (5 wins), recall (4 wins), F1 score (5 wins), and BACC (5 wins) than Shannon entropy in five datasets, falling short only on D6, with a small difference of 0.0002. Analyzing each metric individually, we observed that the best Tsallis parameters resulted in an F1 score gain compared to Shannon entropy of 5.29% and 1.81% in D4 and D5, respectively. Other gains were repeated in ACC, recall, and BACC. In the overall average, our proposal achieved improvements of 0.51%, 1.52%, 1.34%, and 0.62% in ACC, recall, F1 score, and BACC, respectively. Despite a lower accuracy in D3 and D4, this approach alone delivered a BACC of 0.6342 and 0.5845, i.e., it is a supplementary methodology to combine with other feature extraction techniques available in the literature. Based on this, we can state that Tsallis entropy is as robust as Shannon entropy for extracting information from biological sequences.

### 5.4. Case Study IV—Comparing Generalized Entropies

According to the Tsallis entropy results, wherein it overcame Shannon entropy, we realized the strong performance of generalized entropy as a feature descriptor for biological sequences. For this reason, we also evaluated the influence of another form of generalized entropy, such as Rényi entropy [42], as a good feature descriptor for biological sequences. Here, we investigated the performance of Tsallis and Rényi entropy, changing the entropic index for D1, D2, and D3. Moreover, we have chosen the best classifier from case study I (CatBoost).

When considering the same reproducible environment for the experiment, the performance peak was the same for both methods, as we can see in Figure 2, with graphs containing accuracy performance results for all the entropic index values (from 0.1 to 10.0). Regarding the best classification performance, for D1 (Figure 2a), we had ACC: 0.9600, recall: 0.9667, F1 score: 0.9603, and BACC: 0.9600; for D2 (Figure 2b), we obtained ACC: 0.8300, recall: 0.7733, F1 score: 0.8198, and BACC: 0.8300; and for D3 (Figure 2c), we had ACC: 0.7521, recall: 0.359, F1 score: 0.4828, and BACC: 0.649. As seen earlier, Tsallis entropy performs poorly from a specific entropy index onwards, but Rényi entropy demonstrates more consistent performance when compared to Tsallis, representing a possible alternative. Nevertheless, the results again highlight the promising use of generalized entropies as a feature extraction approach for biological sequences.

### 5.5. Case Study V—Dimensionality Reduction

In this last case study, we compared our proposal with other known techniques for feature extraction and dimensionality reduction in the literature, using the same representation of the biological sequences, the k−mer frequency. In particular, for each DNA/RNA sequence, we generated k−mers from k=1 to k=10, while, for proteins, we generated it until k=5, considering the high number of combinations with amino acids. All datasets used have around 1000 biological sequences, considering the prohibitive computational cost to deal with the k−mer approach. In this study, our objective was to use SVD and UMAP to reduce the dimensionality of the k−mer feature vector by extracting new features, as we did in our approach. However, high values of *k* present high computational costs, due to the amount of generated features, e.g., k=6 in DNA (4096 features) and k=3 in protein (8000 features).

From previous case studies, we realized that the feature extraction with Tsallis entropy provided interesting results. Thereby, we extended our study, applying SVD and UMAP in the datasets with k−mer frequencies, reducing them to 24 components, comparable to the dimensions generated in our studies. Fundamentally, UMAP can deal with sparse data, as can SVD, which is known for its efficiency in dealing with this type of data [56,57,58]. Both reduction methods can be used in the context of working with high-dimensional data. Although UMAP is widely used for visualization [59,60], the reduction method can be used for feature extraction, which is part of an ML pipeline [61]. UMAP can also be used with raw data, without needing to adopt another reduction technique before using it [58]. We induced the CatBoost classifier using 10-fold cross-validation. We obtained the results listed in Table 7.

As can be seen, Tsallis entropy achieved five wins, against two for SVD and zero for UMAP, taking into account the ACC. In addition, in the general average, we obtained a gain of more than 18% in relation to SVD and UMAP in ACC, indicating that our approach can be potentially representative for collecting information in fewer dimensions for sequence classification problems.

## 6. Conclusions

In this study, we evaluated the Tsallis entropy as a feature extraction technique, where we considered five case studies with nine benchmark datasets of sequence classification problems, as follows: (1) we assessed the Tsallis entropy and the effect of the entropic index; (2) we used the best parameters on new datasets; (3–4) we validated our study, using the Shannon and Rényi entropy as a baseline; and (5) we compared Tsallis entropy with other feature extraction techniques based on dimensionality reduction. In all case studies, we found that our proposal is robust for extracting information from biological sequences. Furthermore, the Tsallis entropy’s performance is strongly associated with the length of sequences, providing better results when applied in longer sequences. The experiments also showed that Tsallis entropy is robust when compared to Shannon entropy. Regarding the limitations, we found that the entropic index (*q*) affects the performance of ML models, particularly when poorly parameterized. Finally, we highlighted good performance for the entropic index with *q* values between 1.1 and 5.0.

## Figures and Tables

**Figure 1 entropy-24-01398-f001:**
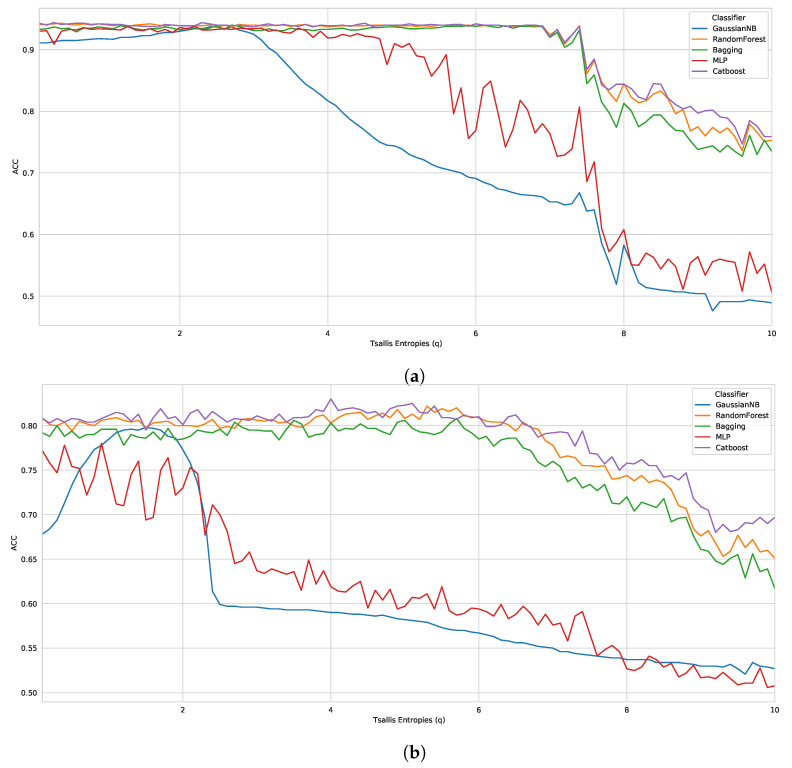
Performance analysis with five classifiers on 100 *q* parameters of three benchmark datasets (evaluation metric: ACC). (**a**) Benchmark D1—ACC; (**b**) Benchmark D2—ACC; (**c**) Benchmark D3—ACC.

**Figure 2 entropy-24-01398-f002:**
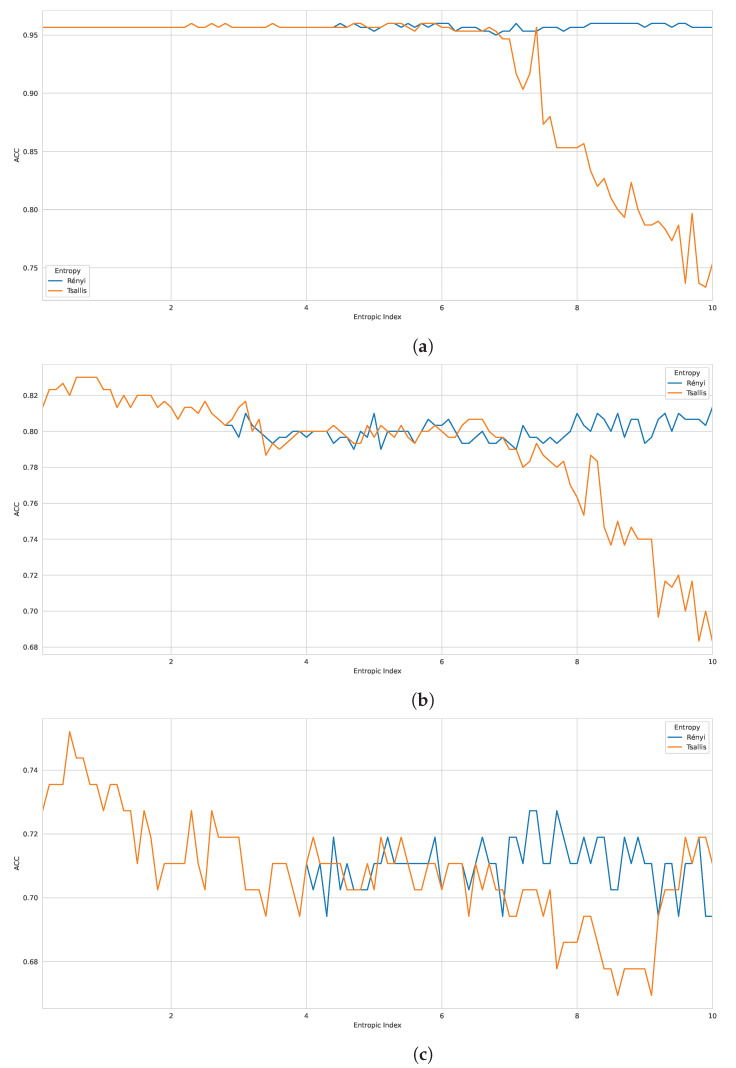
Performance analysis with generalized entropies on 100 *q* parameters of three benchmark datasets (evaluation metric: ACC). (**a**) Benchmark D1—ACC; (**b**) Benchmark D2—ACC; (**c**) Benchmark D3—ACC.

**Table 1 entropy-24-01398-t001:** Feature descriptors found in all studies.

Group	Descriptor
Nucleic Acid Composition	Nucleotide composition
	Dinucleotide composition
	Trinucleotide composition
	Tetranucleotide composition
	Pentanucleotide composition
	Hexanucleotide composition
	Basic k-mer
	Reverse complementary k-mer
	Increment in diversity
	Mismatch
	Subsequence
	GC-content
	AT/GT Ratio
	Cumulative skew
	kGap
	Position-specific nucleotide frequency
	Nucleotide content
	Conformational properties
	Enhanced nucleic acid composition
	Composition of k-spaced Nucleic Acid Pairs
TD	Topological descriptors
K-Nearest Neighbor	K-nearest neighbor for proteins
Autocorrelation	Normalized Moreau–Broto
	Moran
	Geary
	Dinucleotide-based auto-covariance
	Dinucleotide-based cross-covariance
	Dinucleotide-based auto-cross-covariance
	Trinucleotide-based auto-covariance
	Trinucleotide-based cross-covariance
	Trinucleotide-based auto-cross-covariance
Pseudo Nucleic Acid Composition	Type 1 Pseudo k-tuple nucleotide composition
	Type 2 Pseudo k-tuple nucleotide composition
	Pseudo k-tuple nucleotide composition
	Pseudo dinucleotide composition
Numerical Mapping	Z-curve theory
	Nucleotide Chemical Property
	Accumulated Nucleotide Frequency
	Electron–ion interaction pseudopotential
	Pseudo electron–ion interaction pseudopotential
	Binary
	Orthonormal encoding
	Basic one-hot
	6-dimension one-hot method
Amino Acid Composition	Amino acid composition
	Dipeptide composition
	Tripeptide composition
	Terminal end amino acid count
	Amino acid pair
	Secondary structure composition
	Secondary structure—amino acid composition
	Solvent accessibility composition
	Solvent accessibility—amino acid composition
	Codon composition
	Protein length
	Overlapping k-mers
	Information-based statistics
	Basic k-mer
	Distance-based residue
	Distance pair
	Residue-Couple Model
	Composition moment vector
	Enhanced amino acid composition
	Composition of k-spaced amino acid pairs
	Dipeptide deviation from expected mean
	Grouped amino acid composition
	Enhanced grouped amino acid composition
	Composition of k-spaced amino acid group pairs
	Grouped dipeptide composition
	Grouped tripeptide composition
	kGap
	Position-specific nucleotide frequency
Pseudo-Amino Acid Composition	Type 1 PseAAC
	Type 2 PseAAC
	Dipeptide (or Type 3) PseAAC
	General parallel correlation PseAAC
	General series correlation PseAAC
	Pseudo k-tuple reduced AAC (type 1 to type 16)
CTD	Composition
	Transition
	Distribution
Sequence-Order	Sequence-order-coupling number
	Quasi-sequence-order
Profile-based Features	Signal average
	Signal peak area
	PSSM (Position-Specific Scoring Matrix) profile
	Profile-based physicochemical distance
	Distance-based top-n-gram
	Top-n-gram
	Sequence conservation score
	Frequency profile matrix
Conjoint Triad	Conjoint Triad
	Conjoint k-spaced triad
Proteochemometric Descriptors	Principal component analysis
	Principal component analysis (2D and 3D)
	Factor analysis
	Factor analysis (2D and 3D)
	Multidimensional scaling
	Multidimensional scaling (2D and 3D)
	BLOSUM and PAM matrix-derived
	Biophysical quantitative properties
	Amino acid properties
	Molecular descriptors
Sequence Similarity	Gene Ontology (GO) similarity
	Sequence Alignment
	BLAST matrix
Structure Composition	Secondary structure
	Solvent accessible surface area
	Secondary structure binary
	Disorder
	Disorder content
	Disorder binary
	Torsional angles
	DNA shape features
Physicochemical Property	AAindex
	Z-scale
	Physicochemical n-Grams
	Dinucleotide physicochemical
	Trinucleotide physicochemical

**Table 2 entropy-24-01398-t002:** The best and worst parameter (*q*) of each benchmark dataset and classifier, taking into account the ACC metric.

Dataset	GaussianNB		RF		Bagging		MLP		CatBoost	
	*q*	ACC	*q*	ACC	*q*	ACC	*q*	ACC	*q*	ACC
D1	2.7	0.9370	0.4	0.9430	2.7	0.9400	2.2	0.9380	**2.3**	**0.9440**
	9.2	0.4760	9.6	0.7360	9.6	0.7270	10.0	0.5060	9.6	0.747
D2	1.5	0.7980	5.3	0.8220	5.7	0.8080	0.9	0.7800	**4.0**	**0.8300**
	9.6	0.5210	10.0	0.6510	10.0	0.6170	9.9	0.5060	9.2	0.6800
D3	8.7	0.7008	7.8	0.6910	2.0	0.7157	1.5	0.7184	**1.1**	**0.7282**
	1.3	0.6062	9.8	0.5985	9.5	0.5962	0.1	0.6860	5.7	0.6610

**Table 3 entropy-24-01398-t003:** Performance with different entropic index (*q*) values for the sigma70 promoter classification problem.

Dataset	*q*	Classifier	ACC	Recall	F1 Score	AUC	BACC
**D4**	**0.5**	RF	0.6594	0.2556	0.3423	0.6279	0.5647
		CatBoost	0.6563	0.1973	0.2848	0.6233	0.5487
	**2.0**	**RF**	**0.6687**	**0.3094**	**0.3932**	**0.6108**	**0.5845**
		CatBoost	0.6641	0.2063	0.2987	0.6301	0.5567
	**3.0**	RF	0.6672	0.3049	0.3886	0.6150	0.5822
		CatBoost	0.6625	0.2377	0.3282	0.6319	0.5629
	**4.0**	RF	0.6641	0.2825	0.3684	0.6163	0.5746
		CatBoost	0.6656	0.2466	0.3385	0.6415	0.5674
	**5.0**	RF	0.6641	0.2825	0.3684	0.6348	0.5746
		CatBoost	0.6734	0.2646	0.3598	0.6375	0.5775

**Table 4 entropy-24-01398-t004:** Performance with different entropic index (*q*) values for the anticancer peptide classification problem.

Dataset	*q*	Classifier	ACC	Recall	F1 Score	AUC	BACC
**D5**	**0.5**	RF	0.7019	0.5952	0.6173	0.7437	0.6847
		CatBoost	0.6923	0.3810	0.5000	0.7488	0.6421
	**2.0**	RF	0.7019	0.5476	0.5974	0.7454	0.6770
		CatBoost	0.6538	0.4286	0.5000	0.7500	0.6175
	**3.0**	**RF**	**0.7212**	**0.5714**	**0.6234**	**0.7748**	**0.6970**
		CatBoost	0.6827	0.4286	0.5217	0.7385	0.6417
	**4.0**	RF	0.7019	0.5238	0.5867	0.7823	0.6732
		CatBoost	0.6923	0.4762	0.5556	0.7642	0.6575
	**5.0**	RF	0.7211	0.5476	0.6133	0.7813	0.6932
		CatBoost	0.6923	0.4762	0.5556	0.7600	0.6575

**Table 5 entropy-24-01398-t005:** Performance with different entropic index (*q*) values for the SARS-CoV-2 (COVID-19) classification problem.

Dataset	*q*	Classifier	ACC	Recall	F1 Score	AUC	BACC
**D6**	**0.5**	RF	0.9989	0.9992	0.9994	1.0000	0.9985
		CatBoost	0.9982	1.0000	0.9990	0.9999	0.9947
	**2.0**	RF	0.9996	1.0000	0.9998	1.0000	0.9990
		CatBoost	0.9951	0.9996	0.9971	1.0000	0.9862
	**3.0**	**RF**	**1.0000**	**1.0000**	**1.0000**	**1.0000**	**1.0000**
		CatBoost	0.9996	1.0000	0.9998	1.0000	0.9990
	**4.0**	**RF**	**1.0000**	**1.0000**	**1.0000**	**1.0000**	**1.0000**
		CatBoost	0.9996	1.0000	0.9998	1.0000	0.9990
	**5.0**	**RF**	**1.0000**	**1.0000**	**1.0000**	**1.0000**	**1.0000**
		**CatBoost**	**1.0000**	**1.0000**	**1.0000**	**1.0000**	**1.0000**

**Table 6 entropy-24-01398-t006:** Performance of the proposed approach (Tsallis) vs. Shannon entropy (best results in bold). A tie counts one win for each approach.

Dataset	Classifier	Entropy	*q*	ACC	Recall	F1 Score	BACC
D1	CatBoost	Tsallis	2.3	**0.9420**	**0.9673**	**0.9437**	**0.9421**
		Shannon	-	**0.9420**	0.9651	0.9435	**0.9421**
D2	CatBoost	Tsallis	4.0	**0.8140**	**0.7760**	**0.8053**	**0.8153**
		Shannon	-	0.8080	0.7582	0.7970	0.8115
D3	CatBoost	Tsallis	1.1	**0.7231**	0.3869	**0.4724**	**0.6342**
		Shannon	-	0.7207	**0.3886**	0.4708	0.6334
D4	RF	Tsallis	2.0	**0.6687**	**0.3094**	**0.3932**	**0.5845**
		Shannon	-	0.6563	0.2556	0.3403	0.5623
D5	RF	Tsallis	3.0	**0.7212**	**0.5714**	**0.6234**	**0.6970**
		Shannon	-	0.7115	0.5476	0.6053	0.6851
D6	RF	Tsallis	5.0	0.9984	0.9846	0.9915	0.9922
		Shannon	-	**0.9985**	**0.9888**	**0.9922**	**0.9942**
**Mean**	-	Tsallis	-	0.8112	0.6659	0.7049	0.7776
		Shannon	-	0.8061	0.6507	0.6915	0.7714
**Gain**	-	-	-	**0.51%**	**1.52%**	**1.34%**	**0.62%**
**Wins**	-	Tsallis	-	**5**	**4**	**5**	**5**
		Shannon	-	2	2	1	2

**Table 7 entropy-24-01398-t007:** Performance of the proposed approach (Tsallis) vs. SVD vs. UMAP. A tie counts one win for each approach.

Dataset	Reduction	ACC	Recall	F1 Score	BACC
D1	Tsallis (*q* = 2.3)	**0.9430**	0.9650	**0.9438**	**0.9434**
	SVD	0.4980	0.0000	0.0000	0.4982
	UMAP	0.4980	**0.9963**	0.6632	0.4981
D2	Tsallis (*q* = 4.0)	**0.8120**	**0.7718**	**0.8030**	**0.8114**
	SVD	0.5004	0.0016	0.0032	0.5008
	UMAP	0.4994	0.0000	0.0000	0.5000
D3	Tsallis (*q* = 1.1)	**0.7307**	0.3538	0.4541	**0.6310**
	SVD	0.5389	0.7132	**0.4942**	0.5834
	UMAP	0.3191	**0.9933**	0.4825	0.4967
D5	Tsallis (*q* = 3.0)	0.6720	0.5181	0.5515	0.6508
	SVD	**0.7403**	**0.7630**	**0.7752**	**0.7261**
	UMAP	0.4021	0.0000	0.0000	0.5000
D7	Tsallis (*q* = 3.0)	**0.7371**	**0.6711**	**0.6947**	**0.7337**
	SVD	0.5438	0.0000	0.0000	0.4992
	UMAP	0.5143	0.1824	0.1147	0.4963
D8	Tsallis (*q* = 1.1)	0.6500	0.6111	0.6277	0.6525
	SVD	**0.8023**	**0.8575**	**0.7843**	**0.8171**
	UMAP	0.6326	0.7728	0.6544	0.6511
D9	Tsallis (*q* = 9.2)	**0.9489**	**0.9481**	**0.9507**	**0.9481**
	SVD	0.5586	0.6433	0.5517	0.6433
	UMAP	0.5992	0.6528	0.6167	0.6528
**Mean**	Tsallis	0.7848	0.6913	0.7179	0.7673
	SVD	0.5975	0.4255	0.3727	0.6097
	UMAP	0.4950	0.5139	0.3616	0.5421
**Wins**	Tsallis	**5**	**3**	**4**	**5**
	SVD	2	2	3	2
	UMAP	0	2	0	0

## Data Availability

Feature Extraction Technique based on Tsallis Entropy: https://github.com/Bonidia/TsallisEntropy—https://bonidia.github.io/MathFeature/entropy.html accessed on 10 August 2022.

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
