# Peer review of "Information Theory for Biological Sequence Classification: A Novel Feature Extraction Technique Based on Tsallis Entropy"

_entropy, 2022, doi:10.3390/e24101398_

Round 1

Reviewer 1 Report

This is a very excellent work elaborating on the utility of Tsallis and Shannon entropy for biological sequence classification. I only had a few comments:

Major comments:

1. This work introduced a novel and efficient method of biological sequence classification using information entropy. However, too less about biological meaning/interpretations/potential applications were addressed for this novel technique. I suggest adding a paragraph or at least a couple sentences for this method.

2. The Conclusion section is too lengthy -- it should be only a few sentences. I suggest moving some of the contents into the Discussion section.

Minor comments:

1. Line 64: "entropic index" not "Entropic index".

2. Line 139: "long non-coding" not "Long non-coding".

3. Line 148: Expand the abbreviation "SARS-CoV-2" as this was the first time in the text that this term was introduced.

4. Line 155: "assessed" not "assess". 

5. Line 164 and 166: "used" not "use".

6. Line 167: "investigated" not "investigate".

I look forward to seeing a revised version of this work!

Author Response

We added our replies to the comments by reviewer 1 in the attached file.

Reviewer 2 Report

The authors use for comparison of k-mers histograms the Tsallis entropy instead of the more widely used Shannon Entropy.  The experiments are extensive.

Unfortunately, I missed a bit the motivation why the Tsallis entropy and not any other entropy is used. The first part is very short and without new insides here. A more detailed comparison, motivation and also the consideration of other Entropy measurements would be very useful and improve the paper.

Moreover, a deeper literature research to the topic about alignment free comparison of sequences and a classification of the presented work is mandatory.

The experimental results are also ok, but not really show a significant improvement of the use of the Tsallis entropy. A comparison to other entropies and maybe also other information theory-based comparison methods of sequences would be give a better insight. The comparison with SVD/ UMAP I cannot understand, what has this to do with the presented work beside the use of kmers (here exists more sophisticated variants ==> literature research to state of the art and a comparison with this would be better, in my point of few)

Author Response

We replied to all comments by reviewer 2 in the attached file.
